# Exploring Health Research Priority Setting in a South African Province: A Nominal Group Technique Approach

**DOI:** 10.3390/ijerph21070861

**Published:** 2024-06-30

**Authors:** Christo Heunis, Deirdre van Jaarsveldt, Perpetual Chikobvu, Gladys Kigozi-Male, Moroesi Litheko

**Affiliations:** 1Centre for Health Systems Research & Development, University of the Free State, Bloemfontein 9300, South Africa; kigozign@ufs.ac.za; 2School of Nursing, University of the Free State, Bloemfontein 9300, South Africa; vjaarsvd@ufs.ac.za; 3Free State Department of Health, Bloemfontein 9300, South Africa; chikobvup@fshealth.gov.za (P.C.); lithekom@fshealth.gov.za (M.L.); 4Department of Community Health, University of the Free State, Bloemfontein 9300, South Africa

**Keywords:** health research priority setting, online nominal group technique, Free State province

## Abstract

In August 2022, the Free State Provincial Health Research Committee used the online nominal group technique (NGT) for Health Research Priority Setting (HRPS) for the Free State Department of Health (FSDoH) and the research community, considering various stakeholders’ perspectives. This paper explores and describes the identified health research priorities. It also assesses their alignment with the National Health Research Strategy. Additionally, it provides an opinion on the feasibility of using the online NGT for collaborative co-creation of provincial-level health research priorities. Most of the identified health research priorities resonate with the national health research priorities identified by the National Health Research Committee. However, research to “*strengthen surveillance*” was uniquely perceived to be a priority by the participants in the Free State HRPS exercise. A plausible reason for this might be their heightened awareness of the vital role optimal surveillance systems play in coordinating intersectoral responses to pandemics, particularly considering the serious challenges emerging during the initial stages of the COVID-19 outbreak.

## 1. Introduction

Over the past few decades, health research priority setting (HRPS) has emerged as a pivotal area of focus [1,2,3,4]. More specifically, there has been rising interest in the co-creation of research priorities for public health [5] and government-funded applied health research [6]. The co-creation of health research priorities offers stakeholders a unique platform to convene, leverage the latest evidence and criteria, and meticulously identify and rank priority areas. Such processes can ensure that the knowledge needs of a country [7] and, in the context of the current exploratory study, a province, are comprehensively addressed and prioritised.

In South Africa, the 1997 White Paper on Health Transformation [8] initially mandated the development of a research agenda to tackle significant national health challenges, involving diverse stakeholders as equal partners in the process. Section 70(2) of the National Health Act 61 of 2003 [9] states that in identifying health research priorities, the National Health Research Committee—and its provincial counterparts, the Provincial Health Research Committees—must have regard to the burden of disease, the cost-effectiveness of interventions aimed at reducing the burden of disease, the availability of human and institutional resources for the implementation of interventions, the health needs of vulnerable groups such as women, older persons, children, and people with disabilities, and the health needs of communities.

The National Health Research Strategy: Research Priorities for South Africa 2021–2024 [10] identified key focus areas for health research linked to current broad priority disease or mortality areas (according to the national burden of disease) with attention to the multifactorial determinants of health, including biological, psychosocial, and behavioural determinants, as well as health systems, political, economic, and market, planetary and environmental, and other factors, thus incorporating the entire public health domain. Considering this national initiative, advocating for province-wide HRPS becomes imperative.

While the National Health Research Strategy outlined health research priorities for the country at large, knowledge about health research priorities for specific provinces is lacking. The need for HRPS in the Free State was also motivated by the urgency to address wide-ranging public healthcare challenges in this setting [11]. Another factor contributing to the necessity of updating the HRPS was the outdated status of the health research priorities, last revised in 2019, predating the emergence of the COVID-19 pandemic. The 2019 HRPS workshop served as an inclusive, multi-institutional, and interdisciplinary health assembly, aimed at examining research priorities across the following domains: communicable diseases (CDs), non-communicable diseases (NCDs), health policy and systems research (HPSR), violence and injury (V&I), maternal and child health (MCH), and electronic/digital (E/DH) health. Building on the experience gained during the initial HRPS workshop, another workshop entailing exploratory online nominal group technique (NGT) sessions to conduct HRPS among several target groups was held in August 2022.

This paper primarily focuses on exploring and describing the health research priorities for the Free State Department of Health (FSDoH) and the research community, considering the perspectives of multiple stakeholders. Additionally, it examines how these priorities align with the National Health Research Strategy. The secondary focus of the paper is to express an opinion on the feasibility of the use of the online Nominal Group Technique (NGT) by Provincial Health Research Committees to co-create and shape health research priorities at the local level.

## 2. Materials and Methods

### 2.1. Setting

Noting the guidance of the National Health Research Strategy that the particular and complex socio-political, economic, and cultural contexts within South Africa must be regarded to ensure appropriate identification of priority concerns, it should be considered that the Free State represents a relatively unique socio-political, economic, and cultural context for health research. One of nine provinces, the Free State accommodates 5.1% of the country’s public health sector-dependent population, of whom more than 80% are African [12] and historically and socioeconomically disadvantaged due to apartheid spatial and homeland planning [13], inequality in public funding allocation [14], social exclusion, and segregated access to public sector amenities [13,15].

Located on the central plains of the country, the Free State is one of the poorer, more rural provinces. Covering an area of 129,825 km^2^, it represents 10.6% of the total land area of the country [16], and in 2021, it had a population of 2,964,412 [17]. The Free State borders most of the other provinces. To the east, it has an international boundary with Lesotho. With four districts and one metropolitan municipality, the province consists mainly of grasslands, with some Karoo vegetation in the south. It is important to have clear research priorities that reflect the burdens of disease prevalent in the Free State or the central geographical area of South Africa.

Drawn from the Health Systems Trust’s South African Health Review 2022, Table 1 indicates sub-par health indicators for the Free State relative to the country at large [18] (pp. 319–362).

### 2.2. Design

An exploratory, descriptive qualitative research design was implemented to explore and describe health research priorities for the FSDoH and the research community, drawing from the diverse perspectives and experiences of multiple stakeholders [19]. The study capitalised on existing guidelines to conduct HRPS sessions [4,10]. Because health needs in any population constantly change, HRPS sessions require fluidity. Consideration of health research priorities may be difficult due to various important and even somewhat biassed ideas that can emerge [3]. The basic structure of the HRPS session is shaped by the tenacious and competing needs within the health space. The World Health Organisation’s (WHO) (2020) Systematic Approach for Undertaking a Research Priority-Setting Exercise [4] and the National Department of Health’s (NDoH) (2022) National Health Research Strategy [10] provide guidance on how HRPS workshops can be conducted. The WHO [4] (p. 19) states the key aim “is to reach a consensus with a coherent list of priorities rather than a long narrative (shopping list) of everything that needs to be done. This will make communication easier and will encourage uptake and implementation”. The National Health Research Strategy emphasises that the process of setting health research priorities should not be static but rather continuous and cyclical. It should be responsive to the evolving health landscape and local requirements, involving a diverse array of stakeholders. The process should be objective, participative, and strive for consensus. Appropriate preparation and planning are essential. Tools for HRPS include all the resources and instruments required to collect, organise, and analyse the multiple information sources. In the current study, the online NGT [19,20,21] was used.

HRPS is an activity that takes place to decide the key questions or research topics agreed to be priority areas, usually through a process of iterative discussion to reach consensus. This activity assists researchers and policymakers to effectively target research that has the most significant potential and will most benefit the public [1]. In the current study, iterative discussions to reach a consensus on health research priorities for the FSDoH and the research community took the form of online NGT sessions among groups of health managers, providers, and researchers. The research community was defined to include researchers based at the two tertiary institutions in the Free State, namely, the University of the Free State and the Central University of Technology, as well as researchers employed by non-governmental organisations and research councils undertaking research activities in the province at the time of the study.

### 2.3. Data Gathering

The online NGT offers “a powerful and inexpensive tool for systematically collecting and analysing expert opinion” [20] (p. 227). Deemed suitable to identify and prioritise health research areas for the public health provider, the FSDoH, and the Provincial Health Research Committee, this technique was used during an HRPS exercise conducted in August 2022. Table 2 displays the distribution of the 122 individuals who registered for the HRPS workshop by profession and employer. The registrants included 63 healthcare managers from the public sector, 29 health researchers from universities, 12 domain experts from various organisations (5 from universities and 7 from various other sectors, including 3 from non-governmental organisations), 1 health service provider from the private sector, 1 health researcher from a research council, 1 from the public sector (District Clinical Specialist Team), and 1 with affiliations in both the public and university domains. Nineteen healthcare providers from the public sector also registered for the workshop.

Out of the 122 workshop registrants, 22 were breakaway session support staff (comprising information and communication technology [technical] support) and scribes, 7 served as facilitators (one per breakaway session), and 12 were designated as domain experts (2 per breakaway session). Moreover, 12 dignitaries, primarily from the FSDoH, were present at the plenary session. A total of 69 participants actively participated in the NGT breakaway sessions. Among them, 48 cast their votes accurately, while 21 did not, leading to the exclusion of their votes (refer to Table 3).

The NGT is a consensus-building technique whereby group discussions are structured to rank ideas generated in response to a single research question [22,23]. In the current study, after an initial general plenary session where the purpose and method of the NGT were explained and discussed, the respective groups met in separate breakaway sessions or rooms on the electronic platform, each with a facilitator, two domain experts, a scribe, and an information and communication technology support person. The following four steps were followed by each group: (1) silent generation of ideas; (2) round-robin discussion; (3) time-limited discussion; and (4) voting and ranking [19,24].

The silent generation of ideas was the opening step, during which the facilitator posed an open-ended question orally and in writing. A time of silence was then provided for participants to individually generate and write down their ideas. Next, the participants were engaged in a round-robin discussion during which each person shared one idea from their list. At this stage, participants were asked not to interrupt or ask questions, thus strengthening individual voices. The facilitator listed each of the responses on a screened document. Participants were asked to validate the accuracy of the captured data. Additionally, those who followed the sequential process were requested to refrain from duplicating responses until group discussion. Through this approach, participants engaged in member-checking and data winnowing (reduction) autonomously, contributing to the refinement and condensation of the data. The round-robin continued until data saturation was reached. During the time-limited discussion, the list of items was presented, and participants were invited to comment, ask questions, and elaborate on the shared ideas without judgement or criticism. The synergy, or constructive collaboration, created during the discussion stimulated the generation of new ideas. Similar ideas were grouped together. Amendments were made as indicated by the groups. Finally, voting followed to rank the listed ideas. Each participant was requested to choose their top five items and to arrange them in order of importance or priority. Scores ranged from five (5) points for the most important priority to one (1) point for the least important priority. The individuals’ scores were collated on a tally sheet, calculated, and the overall priorities identified.

As suggested by Botma et al. (2010) [25], a trial run of the data collection technique was undertaken. A pilot online NGT session was conducted on Microsoft Teams^TM^ (Microsoft, New York, NY, USA) with the involvement of experienced facilitators, an information and communication technology team, scribes, and members of the research community. A short practice run of the NGT process was facilitated to test the round-robin sharing of ideas, data capturing, and voting by means of Microsoft Forms^TM^. The answers generated were deemed appropriate, and the online functionalities were feasible. Academics well vested in NGT facilitation were used for the actual NGT sessions. Two sessions to train the online facilitators and scribes on the NGT were held to ensure consistency, supported by a written briefing document. Recognising the National Health Research Strategy’s [10] (p. 3) guideline that HRPS should take cognisance of broad priority disease or mortality areas, Table 1 was provided to participants when they registered to allow them to prepare for the NGT sessions.

A principle-based approach to ethics was taken by adhering to the Belmont principles of beneficence, respect for human dignity, and justice [26,27]. Ethical clearance of the research was obtained from the Health Sciences Research Ethics Committee of the University of the Free State.

### 2.4. Data Analysis

Demographic information underwent descriptive analysis, whereas the nominal group data were analysed in two phases: In the first phase of data analysis, conducted during the actual NGT session, participants collaboratively refined, verified, and ranked the shared research priorities across each focus area. The top six priority research topics were identified and highlighted at the top of the listed items from each NGT session, preserving the comprehensive list from each group. Subsequently, the researchers undertook the second phase of data analysis, contextualising the findings within the framework of provincial disease burdens and health system characteristics (refer to Table 1). The full lists of scored items from each of the respective NGT sessions were included in the analysis. In accordance with Rice et al. (2018) [23] (p. 3), this iterative data analysis process started with cleaning the data by removing duplicates or vague items from the lists and renaming similar items. Qualitative content analysis was then performed to code and categorise each dataset [28].

Six steps, described by van Breda (2005) [29] (pp. 4–11), were applied to combine the findings of the respective analyses: Firstly, data were digitally recorded on a computer. Individual datasets were then transferred to electronic spreadsheets, with each set allocated to a row featuring columns indicating the group, code, categories, and sub-categories. Additionally, columns were designated to capture items discussed during the NGT sessions, along with tallied scores per item. Only items receiving a score or a priority score, if attained, were included in the spreadsheet. Secondly, the top five priorities identified by each group were listed in order of priority. Thirdly, a content analysis of the data was conducted. Categories across groups were then identified in relation to the provincial burden of disease and health system characteristics (Table 1). This categorisation involved deductive reasoning, as described by Fisher et al. (2021) [30] (p. 4). Fourthly, confirmation of the content analysis took place. The researchers conducted several brainstorm sessions to finalise the categorisation. Fifthly, the combined ranks were determined. The overall themes consequently emerged as the total scores of the final categories were calculated. Sixthly, a report on the findings from the HRPS process [31] was shared with the Member of the Executive of the FSDoH and various faculties of the two above-mentioned universities, as well as the National Health Research Committee. The findings from the HRPS exercise were also presented and discussed at the Annual Provincial Health Research Day [32].

## 3. Results

Table 4 presents a rank-ordered compilation of research priority themes, accompanied by the respective NGT sessions or originating groups indicated in the final column.

## 4. Discussion

In the present study, the first health research priority identified by online NGT participants in response to the question regarding research priorities for the FSDoH and the research community was the need for research to “strengthen surveillance”. This priority does not align with the priorities outlined in the National Health Research Strategy [10] (pp. 25–26). It is plausible to propose that by the time the NGT sessions occurred in the Free State (August 2022), participants might have acknowledged the pivotal role that robust surveillance systems played during the COVID-19 pandemic. This increased awareness likely originated from participants’ realisation of the urgent requirement for effective coordination across diverse sectors such as law enforcement, education, and social development in addressing pandemics such as COVID-19. This is especially relevant given the challenges encountered during the initial phases of the COVID-19 outbreak in the Free State. These challenges included the need for rapid coordination of COVID-19 district multisectoral response structures, digitisation of COVID-19 data management systems, comprehensive training on infection prevention and control, planning for continuity of essential health services, and standardising district COVID-19 responses. All these processes proved difficult due to varying capacity levels, resources, logistics, operations support, etc. These activities had to be conducted hastily in the interest of an effective and efficient pandemic response. Therefore, it is possible that NGT participants in the Free State might have had misgivings about the province’s current surveillance system and capacity. This realisation may have underscored their perception that research aimed at enhancing surveillance to position the province more effectively to respond to future pandemics should be prioritised.

As indicated below in ranked order, all the other health research priorities identified for the provincial healthcare provider and research community broadly align with the priorities outlined in the National Health Research Strategy [10] (pp. 25–26):

Regarding the second research priority identified in the Free State, “disease management”, the Strategy underscores the importance of research to guide both clinical care and public health interventions, including infection prevention control measures and understanding transmission dynamics [10] (p. 13). Regarding the third priority identified in the province, “Electronic/digital Health”, the Strategy places stress on the importance of research on integrating electronic health records to function seamlessly across the primary, secondary, and tertiary levels of care [10] (p. 23). With reference to the fourth Free State priority, “healthcare service delivery”, the Strategy focuses attention on the need for research to inform the management of service delivery, including referral pathways [10] (p. 10). The Strategy specifically points out the need for research on service and benefit packages, the core services needed, and how to expand these [10] (p. 21). As to the fifth Free State priority, “governance and leadership”, the Strategy [10] (p. 9) stresses research on governance and leadership improvement (“oversight, clinical, accountability, policy frameworks coherence and coordination”). It also points out the need for research to address leadership accountability, advocating for action against “incompetence, racism, and illegal behaviour” [10] (p. 21).

In respect of the sixth Free State priority, “burden of violence and injury”, the Strategy points out the need for research on the prevention of violence and injury and to understand the burden of intentional injuries and road traffic injuries [10] (p. 18). Concerning the seventh Free State priority, “human resources for health”, the Strategy underscores the necessity to conduct research on various aspects of human resources for health, encompassing technical skills [10] (p. 9), as well as structures to evaluate performance and ensure accountability for service delivery. Additionally, it calls for a review of workloads in both the public and private sectors, along with recruitment practices [10] (p. 20). Relating to the eighth Free State priority, “effects of COVID-19”, the Strategy prioritises research aimed at supporting the implementation of diagnostics and products to enhance clinical processes. Additionally, it calls for the development of tools and studies to advance understanding of immune responses and immunity [10] (p. 12). Regarding the ninth Free State priority, “financial and physical resources”, the Strategy emphasises the importance of conducting research on public sector financial management systems and processes, focusing on their efficiency and effectiveness [10] (p. 9). Referring to the tenth Free State priority, “burden of disease”, the Strategy highlights the necessity for research on the burden of CDs, NCDs, MCH, trauma and violence, and mental health [10] (p. 5).

Relating to the eleventh Free State priority, “National Health Insurance”, the Strategy identifies the imperative for research at the “individual, household, community, health sector, and governance” levels across various dimensions to establish norms and standards, facilitate rollout processes, examine financing and financial management mechanisms, garner stakeholder buy-in, and assess cost-effectiveness [10] (p. 10). Although “National Health Insurance” appears in the eleventh place on the list of health research priorities identified in the present study, its importance as a research priority has likely been heightened by the recent enactment of National Health Insurance into law. Consequently, the need for research to inform and guide National Health Insurance implementation might have become more pronounced and urgent. As regards the twelfth Free State priority, “collaboration”, the Strategy highlights the need for research on intersectoral collaboration [10] (p. 25).

Concerning the thirteenth Free State priority, “defaulting”, the Strategy identifies the need for research on methods for enhancing adherence [10] (p. 18). Regarding the fourteenth Free State priority, “patient experience”, the Strategy emphasises the need for research on patient experiences to be included and valued as “legitimate forms of data” [10] (p. 23). With reference to the fifteenth Free State priority, “pandemic preparedness and response”, while the Strategy does not explicitly mention pandemic preparedness, it underscores the necessity for research to enhance the overall response to a pandemic such as COVID-19. This includes focusing on epidemiology, clinical management, infection prevention and control (including safeguarding healthcare workers), candidate therapeutics and vaccine research and development, vaccine effectiveness and safety, as well as integrating social science into outbreak responses [10] (pp. 13–17).

With reference to the sixteenth Free State priority, “health and wellness of staff”, the Strategy prioritises research aimed at understanding healthcare worker dissatisfaction, mental health challenges, and burnout [10] (p. 20). Considering the seventeenth Free State priority, “trauma, rehabilitative and palliative care systems”, the Strategy highlights trauma [10] (p. 17) and emphasises the necessity for research on how rehabilitation and palliative care will be addressed under National Health Insurance [10] (p. 10). Regarding the eighteenth Free State priority, “community education and engagement”, the Strategy identifies the need for research on community engagement and empowerment to optimise community-based care [10] (p. 9). Relating to the nineteenth Free State priority, “environment and climate change”, the Strategy highlights the need for research on environmental factors such as climate change, pollution, and waste management systems [10] (p. 8). When it comes to the twentieth Free State priority, “child health”, the Strategy stresses research to generate a better understanding of neonatal infections, ascertain the HIV profile in children < 5 years, determine the impact of vaccines, particularly on diarrhoea and acute respiratory infections in children < 5 years, and determine why 40% of deaths occur outside of healthcare facilities [10] (p. 19).

Formally mandated bodies such as the National Health Research Committee can provide the vehicle for updating national health research priorities [33], and, as shown in the current exploratory study, Provincial Health Research Committees can do the same at the provincial level. The current research suggests that the application of an established HRPS tool, such as the NGT, provides a feasible and inexpensive means for these committees to identify and rank health research priorities to ensure that health research aligns with local and community needs.

As established in a systematic review of how HRPS is conducted in low- and middle-income countries, challenges in HRPS include engagement with stakeholders, data availability, and capacity constraints [2]. Furthermore, in the absence of published information on implementation or evaluation, it is not possible to assess what the impact and effectiveness of HRPS may have been. With reference to the current HRPS exercise, it will be necessary to not only regularly repeat the process to set health research priorities but also to monitor their actual impact in generating relevant health research for the public health provider and research community in the Free State.

Judging from the topics of research proposals serving for FSDoH approval since the HRPS was provided to the two universities in the Free State in March 2023, there has already been an increase in research related to “strengthening surveillance”. Early indications suggest that the provided list of health research priorities is actively being considered and is positively influencing the direction of research towards the identified priority areas. However, for future research, a more rigorous evaluation of the implementation (uptake) of the health research priority areas is recommended.

When discussing the limitations of the research, it is crucial to acknowledge that findings from HRPS exercises often lack conventional scientific validity or representativeness. Instead, they serve as preliminary guides to identify knowledge needs, informing the priorities of individual institutions and funding bodies [33].

The highly structured format of the NGT is a potential drawback, as it could constrain participants who wish to share their narratives more extensively [34]. This limitation is exacerbated in online data collection settings, where lengthy responses can lead to disengagement from those listening. NGT facilitators in the current study also encountered the challenge of maintaining participant focus and involvement. Strategies like circular sharing were employed to ensure everyone had an equal chance to contribute. Additionally, due to voting errors, a considerable number of the votes had to be discarded during the analysis.

The exclusion of patients may have restricted the diversity of viewpoints in the present study. Integrating both patients’ and providers’ perspectives could enhance the breadth and convergence of opinions [35]. Furthermore, excessive heterogeneity within the groups may have impeded the achievement of consensus [20]. Group dynamics and free participation can be hindered when all members are new to the team, as was the case in the current study, and when power differentials arise [36]. However, appropriate use of the NGT mitigates the risk of bias from dominant participants by offering all participants an equal opportunity to express their views in a round-robin fashion and employing ranking to identify high-priority research areas [37].

The experience of conducting HRPS online in the current study aligns with previous research utilising the NGT methodology in HRPS. For instance, Mason et al. (2021) [19] utilised online NGT discussions to gather opinions from palliative care clinicians and prioritise their understanding and needs regarding clinical research in Greece, Romania, and Spain. These authors concluded that conducting NGT meetings online was feasible and potentially advantageous compared to traditional face-to-face meetings, although it required thorough preparation for effective participant contribution. The validity of these assertions was supported by the current study, which capitalised on the ability of online NGT to engage participants in distributed locations and, because of its time-limited nature, allowed for the participation of healthcare managers and providers with busy schedules. The current study also emphasised the importance of timeously mobilising health information and communication technology resources, expertise, training, and tools necessary for collecting, organising, and analysing information from online NGT sessions. Similarly, following the online NGT methodology, Kulczycki and Shewchuk (2008) [20] utilised a novel web-based interface to systematically gather and prioritise responses regarding encouraging providers to recommend diaphragm use. Their study also validated the use of the online NGT as a viable method for aggregating expert opinions in areas with limited evidence.

## 5. Conclusions

This paper describes the process and results of an online NGT exercise conducted by the Free State Provincial Health Research Committee to determine priority areas for health research for the provincial health department and research community. Consistent with prior research utilising the online NGT for HRPS, this methodology proved feasible in the co-creation of research priorities in the realm of health. Insights were gathered from healthcare managers, service providers, and researchers. From the tallied results, the top five health research priorities identified included (1) “strengthening surveillance”, (2) “disease management”, (3) “Electronic/digital Health”, (4) “healthcare service delivery”, and (5) “governance and leadership”. Of the full list of 20 identified priorities, 19 resonate with the national health research priorities identified by the National Health Research Committee. However, it appears that during the COVID-19 pandemic, there was a distinct emphasis on prioritising research to “strengthen surveillance” in the Free State. This emphasis might have been influenced by the timing of the HRPS event in August 2022, coinciding with increased recognition of the crucial role surveillance plays during pandemics.

## Figures and Tables

**Table 1 ijerph-21-00861-t001:** Health indicators reflecting sub-par healthcare performance in the Free State.

Indicator			Year/s	FS	SA
**Mortality**	Life expectancy at birth (years)	Female	2016-21	61.4	64.6
		Male	2016-21	55.5	59.3
	Excess death rate per 100,000 population		2020/21	526	456
	Maternal mortality in facility ratio per 100,000 live births		2020/21	178.8	120.9
	Early neonatal death in facility rate		2020/21	11.9	9.7
	Neonatal death in facility rate		2020/21	16.0	12.6
	Perinatal death in facility rate		2020/21	38.7	29.8
	Stillbirth in facility rate per 1000 total births		2020/21	27.1	19.9
**Communicable disease**	TB	DSTB death rate (%)	2019	10.6	7.4
		DSTB treatment success rate (%)	2019	77.3	79.3
		MDRTB treatment success rate (%)	2019	59.1	60.7
		XDR treatment success rate (%)	2019	46.2	49.9
	HIV	HIV prevalence (total population) (%)	2021	14.0	13.4
		ART effective coverage (%)	2021	53.8	59.4
**Reproductive health**	Delivery 10-19 years in facility rate (%)		2020/21	13.1	14.3
	Antenatal 1st visit before 20 weeks rate (%)		2020/21	61.3	67.9
	Antenatal 1st visit coverage (%)		2020/21	78.5	83.9
**Child health**	Child < 5 years pneumonia incidence (%)		2020/21	13.1	12.6
	Child < 5 years severe acute malnutrition incidence (%)		2020/21	4.1	1.5
	DTaP-IPV-Hib-HBV 3rd dose coverage (%)		2020/21	79.8	82.7
	Immunisation < 1 year coverage (%)		2020/21	75.9	79.5
	Measles second dose coverage (%)		2020/21	73.3	76.4
	PCV third dose coverage (%)		2020/21	77.6	82.3
	Pneumonia case fatality < 5 years rate (%)		2020/21	3.1	2.1
	RV second dose coverage (%)		2020/21	78.7	83.2
**Chronic disease and risk factors**	Diabetes prevalence (%)		2020	11.7	10.4
	Diabetes treatment coverage (%)		2020	36.0	37.5
	Hypertension prevalence (15 + years) (%)	Both sexes	2017	33.2	28.2
		Females	2017	36.7	29.0
		Males	2017	29.0	27.4
**Injury and risk behaviour**	Prevalence of smoking (%)		2017	19.70	19.30
	Road accident fatalities per 100,000 population		2019	29.5	21.7
**Health service**	PHC utilisation rate < 5 years (average number of visits per person)		2020/21	2.5	2.6
**Health personnel**	Clinical associates per 100,000 population		2021	0.5	0.9
	Dental specialists per 100,000 population		2021	0.0	0.3
	Dental therapists per 100,000 population		2021	0.0	0.7
	Enrolled nurses per 100,000 population		2021	48.6	62.8
	Medical practitioners per 100,000 population		2021	33.3	36.5
	Optometrists per 100,000 population		2021	0.2	0.5
	Professional nurses per 100,000 population		2021	102.3	153.6
	Psychologists per 100,000 population		2021	1.3	1.5
	Speech therapists and audiologists per 100,000 population		2021	0.8	1.7
**Health financing**	Expenditure per patient day equivalent (district hospitals) (R)		2019/20	3040	3179
	Medical scheme coverage (%)		2020	4.0	14.8

Abbreviations: TB, tuberculosis; DSTB, drug-susceptible TB; MDRTB, multidrug-resistant TB; XDRTB, extremely drug-resistant TB; HIV, human immunodeficiency virus; DTaP, diphtheria, tetanus, and pertussis; Hib, haemophilus influenzae type b; HBV, hepatitis B vaccine; PCV, pneumococcal conjugate vaccine; RV, rotavirus vaccine; PHC, primary health care.

**Table 2 ijerph-21-00861-t002:** Distribution of workshop registrants by profession and employer.

Profession	Employer	Number of Registrants
Healthcare manager	Public sector	63
Health researcher	University	29
Healthcare service provider	Public sector	19
Healthcare service provider	Public and private sectors	4
Healthcare service provider	Private sector	1
Health researcher	Non-governmental organisation	3
Health researcher	Research Council	1
Healthcare manager	Public sector (District Clinical Specialist Team)	1
Healthcare manager and researcher	Public and university	1
Total	-	122

**Table 3 ijerph-21-00861-t003:** Distribution of NGT workshop registrants and breakaway session participants.

	Workshop Registrants						NGT Breakaway Session Participants		
Sessions	Number of NGT Participants	Number of Support Staff	Facilitators	Domain Experts	Dignitaries	Total	Number Who Voted Correctly	Number Who Vote Incorrectly	Total Number Who Voted
Plenary session	69	22	7	12	12	122	-	-	-
Breakaway sessions									
CDs (1 & 2)	20	7	2	2	-	31	13	7	20
NCDs	16	3	1	2	-	22	7	9	16
V&I	7	3	1	2	-	13	7	0	7
MCH	9	3	1	2	-	15	7	2	9
E/DH	8	3	1	2	-	14	5	3	8
HPSR	9	3	1	2	-	15	9	0	9
Total	-	-	-	-	-	-	48	21	69

Abbreviations: CDs, communicable diseases; NCDs, non-communicable diseases; V&I, violence and injury; MCH, maternal and child health; E/DH, electronic/digital health; HPSR, health policy and systems research.

**Table 4 ijerph-21-00861-t004:** Identified health research priority themes.

Identified Research Priority Themes	Total Score	NGT Session
1. Strengthening surveillance	108.5	CD1; CD2; NCD; MCH; E/DH; HPSR
2. Disease management	94	CD1, CD2, HPSR, MCH, NCD, V&I
3. Electronic/digital Health	60	E/DH
4. Healthcare service delivery	58	HPSR, MCH, NCD
5. Governance and leadership	50.5	HPSR, MCH, NCD
6. Burden of violence and injury	48	V&I
7. Human resources for health	40	HPSR, MCH, NCD, V&I
8. Effects of COVID-19	34.5	CD1, CD2
9. Financial and physical resources	34	CD, HPSR, MCH, NCD, V&I
10. Burden of disease	31	HPSR, NCD
11. National Health Insurance	26	HPSR, NCD
12. Collaboration	22	HPSR, NCD
13. Defaulting	20	CD2, NCD
14. Patient experience	18	CD2, HPSR, NCD, V&I
15. Pandemic preparedness and response	16.5	CD1, HPSR
16. Health and wellness of staff	15	HPSR, V&I
17. Trauma, rehabilitative, and palliative care systems	15	NCD, V&I
18. Community education and engagement	14	CD1, CD2, NCD
19. Environment and climate change	8	CD1, CD2, NCD
20. Child health	7	MCH

Abbreviations: CD, communicable disease; NCD, non-communicable disease; MCH, maternal and child health; E/DH, electronic/digital health; HPSR, health policy and systems research; V&I, violence and injury.

## Data Availability

The datasets presented in this article are not readily available because this would compromise participant confidentiality. Requests to access the datasets should be directed to [Dr D van Jaarsveldt, School of Nursing, University of the Free State, Bloemfontein 9300, South Africa, vjaarsvd@ufs.ac.za].

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
