# Peer review of "Exploring Health Research Priority Setting in a South African Province: A Nominal Group Technique Approach"

_ijerph, 2024, doi:10.3390/ijerph21070861_

Round 1
Reviewer 1 Report
Comments and Suggestions for Authors
Very interesting article. My comments are well taken care of and the manuscirpt is clear now. Congratulations.
Reviewer 2 Report
Comments and Suggestions for Authors
The author/s tried to improve the paper. However, the changes make it even more problematic. If the text is not the case study, what should it been? The goal is very loosely defined: "This paper delineates the identified research priorities" (check the meaning of "delineate" in the vocabulary). As a person which has zero knowledge about the local situation I really do not know, what is the main message from the paper. Is the goal to evaluate the quality of the Free State PHRC HRPS (by checking the importance/relevance of priorities)? Or to document the participatory approach and its importance? What is the value added of related messages for the international reader???
The request for the literature review part and internationalisation of the text was not respected.
Reviewer 3 Report
Comments and Suggestions for Authors
Abstract line 20 suggest deleting Seemingly
Introduction line 43, remove " " and italics
Line 48 remove furthermore
Line 51 remove that
Line 53 items listed do not need to italicized
Table #1, spell out second dose and third dose. It is changing your spacing
Line 86, change from first person we. The to the research was to understand the various perspectives and experiences.
Line 92 remove thus
Line 96 remove (p.19)
Suggest change line 104 In this study, the internet-based.
Line 121 remove (p. 227)
For table #2 Were all of these individuals on one meeting? Why representation of just one person for some of the profession with employer?
Line 136 steps do not need italicized
Line 140 and 145 and 147 remove italic
Provide better descriptions of lines 170-172
Lines 179 and 183 remove italic
Remove secondly and thirdly and fourthly, etc (remove italic) line 197 and 198
Table #3 remove italic
Check with journal editors about lines 219 and on about italics
Line 287 spell out first
Line 314 remove thus
Any limitations to the study as it relates to the establishment of priorities?
Comments on the Quality of English Language
Concerned about all of the italics as well as pp. number throughout paper
Round 2
Reviewer 2 Report
Comments and Suggestions for Authors
By this version I slowly start to understand what authors want to achieve. I hope that I am true:
a/ To describe prorities defined by the NGT and to link them to the NHRS. This should be done by formulating RQ1.
HOwever, the authors can do much more. NGT represents the interesting example of co-creation and should be evaluated from this point of view. I suggest to add RQ2 related to the NGT as the co-creation mechanism. This would allow the framing of the text within the existing literature on the topic co-creation (the core theme of the IASIA conference in July 2024 in South Africa as example).
If this second dimension is added, the paper has the potential to be published.
